# Genome-wide analysis of MADS-box transcription factor gene family in wild emmer wheat (*Triticum turgidum* subsp. *dicoccoides*)

Ghader Mirzaghaderi ⓘ *

Department of Plant Production and Genetics, Faculty of Agriculture, University of Kurdistan, Sanandaj, Iran

* gh.mirzaghaderi@uok.ac.ir

**Data Availability Statement:** All relevant data are within the manuscript and its Supporting Information files.

## Abstract

The members of MADS-box gene family have important roles in regulating the growth and development of plants. MADS-box genes are highly regarded for their potential to enhance grain yield and quality under shifting global conditions. Wild emmer wheat (*Triticum turgidum* subsp. *dicoccoides*) is a progenitor of common wheat and harbors valuable traits for wheat improvement. Here, a total of 117 MADS-box genes were identified in the wild emmer wheat genome and classified to 90 MIKC$^C$, 3 MIKC*, and 24 M-type. Furthermore, a phylogenetic analysis and expression profiling of the emmer wheat MADS-box gene family was presented. Although some MADS-box genes belonging to *SOC1*, *SEP1*, *AGL17*, and *FLC* groups have been expanded in wild emmer wheat, the number of MIKC-type MADS-box genes per subgenome is similar to that of rice and Arabidopsis. On the other hand, M-type genes of wild emmer wheat is less frequent than that of Arabidopsis. Gene expression patterns over different tissues and developmental stages agreed with the subfamily classification of MADS-box genes and was similar to common wheat and rice, indicating their conserved functionality. Some TdMADS-box genes are also differentially expressed under drought stress. The promoter region of each of the TdMADS-box genes harbored 6 to 48 responsive elements, mainly related to light, however hormone, drought, and low-temperature related cis-acting elements were also present. In conclusion, the results provide detailed information about the MADS-box genes of wild emmer wheat. The present work could be useful in the functional genomics efforts toward breeding for agronomically important traits in *T. dicoccoides*.

## Introduction

Wheat is an important crop worldwide, occupying 17% of global cultivated lands and providing 30% of global calorie consumption [1]. However, abiotic stresses, such as drought and salinity, have a significant impact on its yield, particularly under changing climate conditions. Wild emmer wheat (*Triticum turgidum* ssp. *dicoccoides*; common name: *T. dicoccoides*), the progenitor of the A and B genome of bread wheat, has been adapted to abiotic stress during evolution and has a great potential for wheat improvement [2, 3]. Identification of genes

**Funding:** GM was supported by Iran National Science Foundation (INSF) grant 99014038. The funding doesn't include publication fee.

**Competing interests:** The author have declared that no competing interests exist.

associated with stress tolerance in wild emmer wheat, helps us to understand the mechanism underlying stress response which can be applied in wheat breeding programs.

MADS-box genes compose a regulatory family of transcription factors found in all eukaryotes and play a crucial role in controlling various aspects of plant growth and development, including flowering, fruit ripening, and seed formation. MADS-box genes have been well documented in Arabidopsis and rice and have been studied in common wheat [4–6] and many other plants. Genes associated with stress tolerance in wild emmer wheat have been identified [7–10]. It has been shown in model plants that some MADS-box genes modulate tolerance to drought [11–13] and cold [14]. For example, *OsMADS26*-down-regulated rice plants are more tolerant to drought without a strong impact on plant development [15]. There are evidences that the induction of *OsMADS27* mediates salt tolerance in rice [16]. In Arabidopsis, MADS-box genes are involved in response to water stress and drought resistance possibly by the regulation of abscisic acid (ABA) pathway [17]. Beside these evidences of the involvement of the MADS-box genes in plant growth, development and tolerance against stresses, the detailed information on MADS-box gene family is not available yet in wild emmer wheat.

It has been known for decades that the floral homeotic genes, AG (AGAMOUS) from *Arabidopsis thaliana* and DEF A (DEFICIENS A) from snap dragon (*Antirrhinum majus*), share strong sequence similarity with DNA-binding domain of SRF (SERUM RESPONSE FACTOR) transcription factor of humans and MCM1 (MINICHROMOSOME MAINTENANCE 1) of yeast. This conserved domain has since been named the MADS-box followed by the initials of MCM1, AG, DEF, and SRF. Based on the sequence of this highly conserved MADS domain which is a 58–60 amino acid DNA-binding sequence, two types of MADS-box has been distinguished [18]. The first type is known as M-type or type I MADS-box genes, which commonly contain the MADS-box domain without any other conserved domains. The second type is type II or MIKC-type MADS-box genes which harbour MADS-, I-, K-, and C-terminal domains. The additional domains downstream of the MADS-box MIKC-type proteins, especially the conserver keratin-like (K) domain play a role in protein interactions and dimerization [19, 20]. A short intervening (I) domain, separates the MADS and K domains. The I domain may also be involved in interaction with other proteins [21]. MIKC-type MADS-box proteins may also contain a variable C-terminal domain that involves in protein interaction, transcription activation or protein modification [22, 23]. Because the function of the C domain has not been clearly defined due to its variability, MIKC-type MADS-box proteins that have MADS and K domains are considered as fully functional.

The type I MADS-box proteins have been divided into Mα, Mβ, and Mγ clades [24]. In *A. thaliana* some members of the type I genes are important for normal development of the female gametophyte or endosperm and may be responsible for post-zygotic lethality in interspecific hybrids [25–31]. Plant Type II proteins are also divided into MIKC$^C$ and MIKC* [32]. In angiosperms and ferns, various classes of MIKC$^C$ genes have been identified while only two classes of MIKC* genes have been recognized [33, 34] based on phylogenetic relationships. Several studies imply the importance of MIKC* genes in pollen development [35–38]. On the other hand, the genes of MIKC$^C$ class play important roles in flowering time, floral organ identity, and fruit development [39–44]. The role of the MADS-box gene family is not confined to flower development. They are key components of the gene regulatory networks associated with the distinct developmental fates in the root [45] and are involved against various stress conditions [12, 46].

Here, I performed an in-silico genome-wide investigation to identify the MADS-box family members in wild emmer wheat. The phylogenetic relationship, physical localization, gene structure, conserved domain, cis-acting elements, and related micro RNAs (miRNAs) of the identified MADS-box genes were analyzed. Furthermore, the expression patterns of MADS-

box genes in different tissues and time points were investigated using publicly available RNA-seq and microarray data. This study provides information about the important candidate MADS-box genes for further wheat breeding programs.

## Materials and methods

### Identification of MADS-box genes in *T. dicoccoides*

Genomic DNA, protein, and transcript sequences, and the annotation file of *T. dicoccoides* were downloaded from EnsemblPlants (WEWSeq_v.1.0, https://plants.ensembl.org/). The Multiple Sequence Alignment for the MADS-box family was also downloaded from the plant transcription factor database [47] and used to make a Hidden Markov Model (HMM) profile by the HMMER package [48]. The HMM was used as a query to identify the MADS-box proteins of *T. dicoccoides* at the 0.001 p-value cut-off (S1 Table). To differentiate type I (M-type) and type II (MIKC-type) MADS-domain proteins, the *T. dicoccoides* MADS-box protein sequences were aligned with all MADS-box proteins of Arabidopsis [24] and rice [49] with MAFFT (L-INS-i strategy) [50] using just the MADS domain part of the sequences. A phylogenetic tree was constructed using IQTREE [51] and ModelFinder [52].

### Naming of MIKC-type MADS-box genes

The identified MADS-box genes were named as follows: The name of each *T. dicoccoides* MADS-box gene is composed of the 'Td' prefix which refers to *T. dicoccoides*, plus the name of the most similar *Arabidopsis thaliana* (or *Oryza sativa* in case that the gene was not found in Arabidopsis) gene which was inferred from the phylogenetic analysis (see below), their subgenome location (A or B) and subfamily association. Identical gene names were assigned to the putative homoeologs except for the subgenome identifier (e.g. *TdAG-1A* and *TdAG-1B*). Homoeologs were identified by referring to the EnsemblPlant database. Inparalogs (i.e. duplicated copies) were indicated by consecutive numbers separated by a dash so that the name of the gene with the ID TRIDC3AG061490 is *TdFLC-3A-4* as it is the fourth *TdFLC* gene on 3A chromosome (S1 Table).

### Physical characterization of MADS-box proteins

The *T. dicoccoides* annotation file was used to display the structure of the MADS-box genes using the Gene Structure Display Server (GSDS, http://gsds.cbi.pku.edu.cn) [53]. The conserved domains of the MADS-box proteins were identified from the Conserved Domain Database (CDD) [54] web server, and the output file was used to visualize the domain structure of the MADS-box proteins in TBtools [55]. The physical map of the MADS-box genes on *T. dicoccoides* chromosomes was generated using shinyCircos2 [56]. The intron rages and frequencies of MADS-box genes were determined in TBtools (S2 Table).

### Maximum likelihood phylogeny of MADS-box proteins

Based on the first phylogeny mentioned above, MADS-box subfamily sequences of *T. dicoccoides*, Arabidopsis [24] and rice proteins [49] were aligned using MAFFT (E-INS-i strategy). Subfamily alignments were then merged using MAFFT (E-INS-i algorithm) [50]. The resulting alignment was trimmed using the kpi-gappy strategy of the ClipKIT tool [57], and a maximum likelihood tree was inferred using the trimmed alignment with IQTREE [51]. The best amino acid substitution model was determined with the ModelFinder option based on the Bayesian information criterion (BIC) and the JTT+F+G4 was chosen [52] and 1000 ultrafast bootstraps

were applied [58]. the MIKC* subclade was set as the outgroup and the generated Newick tree file was visualized in R using the 'ggtree' package [59].

## Expression of MADS-box genes

153 samples RNA-seq data generated from 20 different combinations of wild emmer wheat (genotype Zavitan) tissues and developmental stages belonging to root, leaf, flag leaf, flower (anthers and carpels), glume, lemma and palea, grain, and different stages of developing spike were downloaded from SRA database of NCBI (Accession: ERP022006) [60]. After quality control and trimming the low-quality section of reads, the read data from each sample were aligned to the *T. dicoccoides* reference genome using HISAT2, and transcripts assembling and merging were done using StringTie with default settings [61]. Normalization of abundance estimates as FPKM (fragments per kilobase of transcript per million mapped reads) values, for the MADS-box genes were extracted using the ballgown package [62]. A heatmap was produced from log2(FPKM+1) (FPKM: fragments per kilobase of transcript per million fragments mapped) values. of MADS-box genes of *T. dicoccoides* over the developmental stages using the 'pheatmap' package. The co-expression of the MADS-box genes were analyzed by clustering using the R package WGCNA [63].

To assess the TdMADS-box gene response to drought stress, I further used microarray data (Gene Expression Omnibus (GEO) dataSets; accession: GSE31762) from a transcriptome analysis of terminal drought response applied at the inflorescence emergence stage [Zadoks 50–60, 64], after emergence of 1–2 spikes, flag leaf samples were analyzed. The microarray data belonged to two drought tolerant (Y12-3) and drought susceptible (A24-39) genotypes different in their yield and yield stability under drought stress [65]. The orthologous genes of *T. dicoccoides* were identified by Blastn of the common wheat cDNA against the *T. dicoccoides* cDNA sequences. Mean expressions were presented based on transcript per million (TPM) as log2(TPM + 1). Mean expression of MADS-box genes between well-watered and terminal drought conditions was compared using t-test and the bar plots of the differentially expressed genes between the two conditions were produced using the 'ggplot2' package [66].

## Cis-regulatory elements of MADS-box genes

The 2-Kb upstream sequences of MADS-box genes were extracted from the *T. dicoccoides* genome using TBtools [55]. The cis-acting elements of the sequences were predicted with the online PlantCARE tool [67].

## MicroRNA (miRNA) target of MADS-box genes

Targeting miRNAs of MADS-box genes of *T. dicoccoides* were predicted using the Analysis page on psRNATarget website v2.0 [68]. Both the cDNA sequences of (corresponding to the longest protein variants) and intronic sequences of the TdMADS-box family were uploaded separately. The default parameters were used except that the expected value was set to 1.5. miRNA targets of cDNA and intronic sequences were separately downloaded and presented in an excel data sheet (S3 Table).

## Results

### Frequency and physical distribution of MADS-box genes in *T. dicoccoides*

Here, a total of 596 transcript variants belonging to 117 MADS-box genes were identified in the wild emmer using the genome assembly WEWSeq_v.1.0 [60]. Only the longest transcript variant from each gene was kept for downstream analysis. The MADS-box genes were named

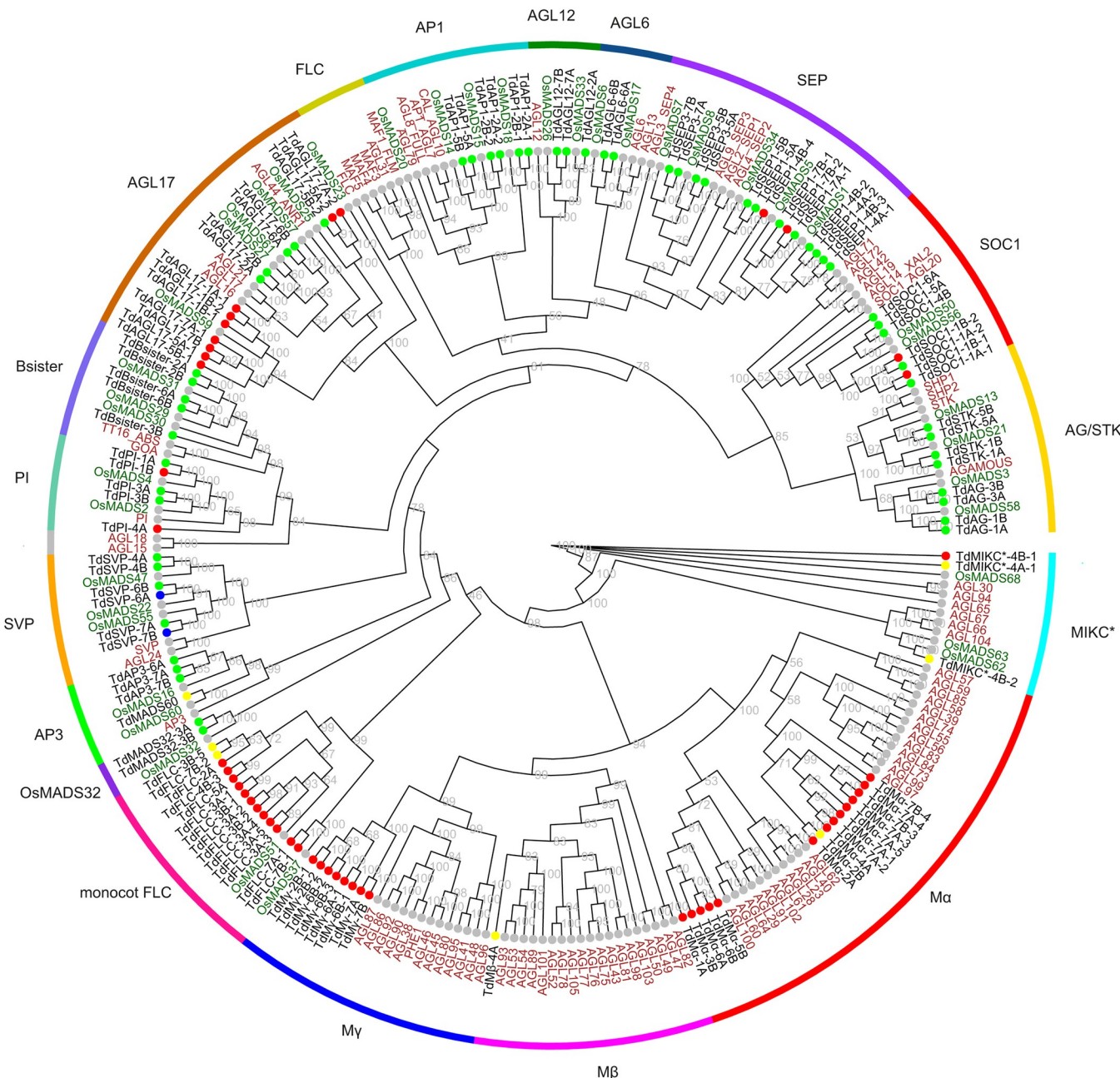

**Fig 1. Maximum likelihood phylogeny of MADS-box proteins from wild emmer wheat (*Triticum dicoccoides*), rice (*Oryza sativa*), and Arabidopsis.** A phylogenetic unrooted tree of MADS-box proteins from *T. dicoccoides*, rice, and Arabidopsis was inferred using MAFFT-aligned sequences and IQ-Tree [51, 52]. *T. dicoccoides* genes are colored black, whereas rice and Arabidopsis genes are in green and red, respectively. Subfamilies are indicated outside the tree. Dots next to *T. dicoccoides* gene names indicate the presence of MADS-box (red), K-box (blue), or both (green) within the coding region of the gene as detected by CDD. Yellow circles: none was detected. Accession numbers of *T. dicoccoides* genes are available in S1 Table.

according to their subfamily relationship (Fig 1 and S1 Table). The corresponding 117 proteins were classified into 3 major groups i.e. 90 MIKC$^C$, 3 MIKC*, and 24 M-type based on phylogenetic results. The maximum number of MADS-box genes were found on chromosome 7A which harbored 14 genes followed by 7B with 12 genes, whereas, each of the other chromosomes had 7 or 8 MADS-box genes. MIKC$^C$-type MADS-box genes were almost randomly

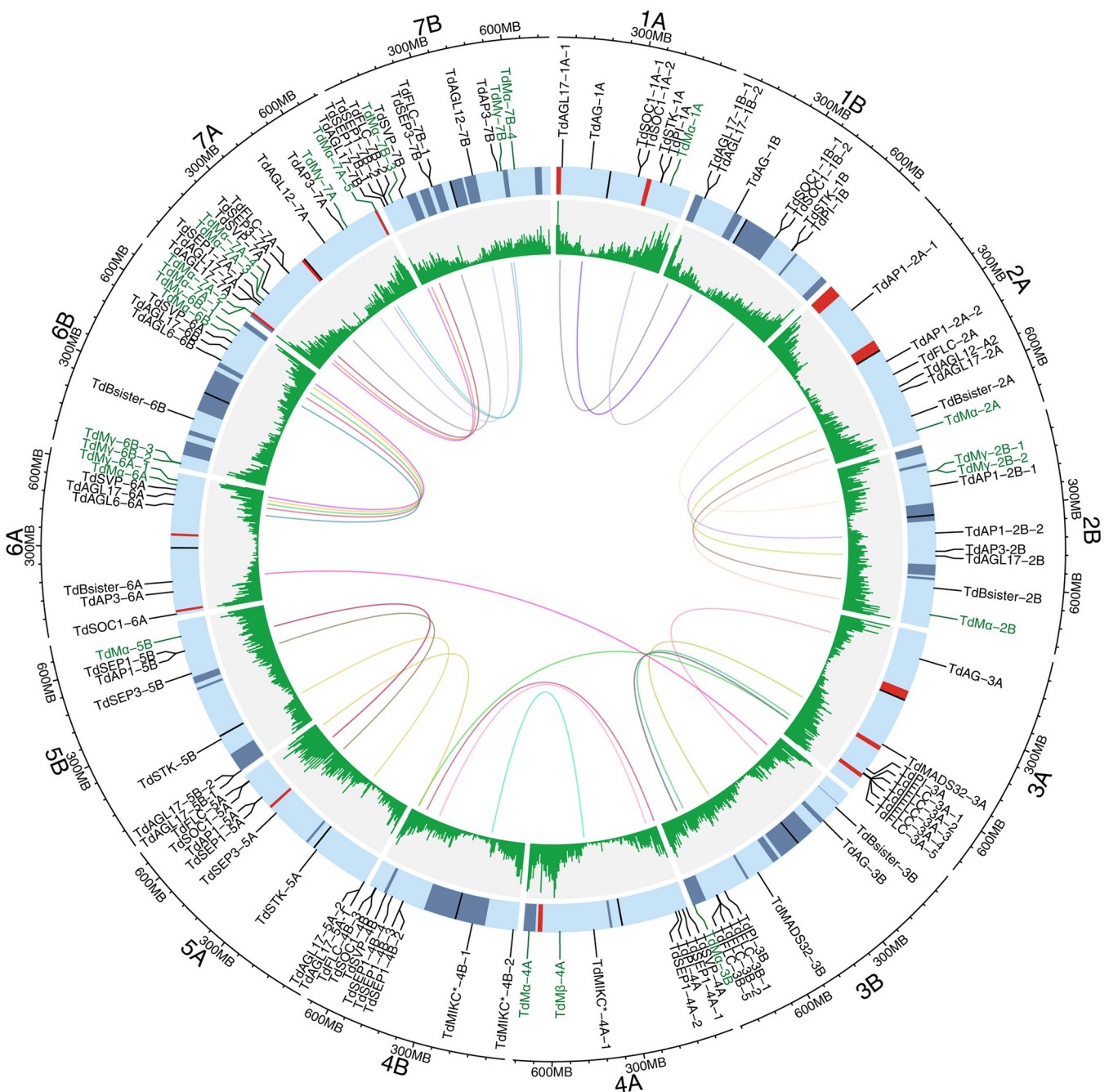

**Fig 2. Chromosomal location of MADS-box genes on *T. dicoccoides* genome.** The genes were mapped to 14 *T. dicoccoides* chromosomes on which the overall gene density heatmap is presented as well. Chromosome numbers are indicated outside the outer circle. Homoeologous genes are connected using central links. Chromosomes are banded according to p*Ta*535-1 (red bands) and (GAA)$_{10}$ (blue bands) FISH patterns. M-type MADS-box genes are highlighted with green color.

distributed on all the chromosomes. The 24 copies of M-type genes of wild emmer wheat were distributed over all the chromosomes except that 11 were predominantly located on homoeologous group 7 (Fig 2). The MIKC* genes along with the only Mβ MADS-box gene are located in homoeologous group 4 (Fig 2). None of the M-type genes contained K domain (Fig 1). As mentioned in Introduction, the functionality of the MIKC-type MADS-box genes is mostly

determined by the presence of MADS and K domains. Among the identified MADS-box genes, 58 encodes both MADS and K domains (49.57%), while 50 genes lacked K domain (42.73%), two lacked MADS box (1.71%) and 7 lacked neither MADS nor K domain based on the CDD results under the applied threshold of 0.05. None of the M-type genes contained K domain (Fig 1 and S1 Fig). Other domains also found in some MADS-box genes including DUF6119 (in *TdFLC-3A-2*), PABP (in *TdMγ-2B-1*), SNAPc (*TdMγ-2B-1*), ARG80 (*TdMα-2A*, *TdMα-7A-3*, *TdMα-7A-4*, *TdMγ-7A* and *TdMα-7B-3*), HD-ZIP (*TdSEP1-5A* and *TdSEP1-5B*), TIM (*TdAP3-2B*), HU_IHF (*TdFLC-7B-1*), KLF8 (*TdMα-7B-4*) and SRP54 (TdPI-4A) were also found.

Gene structure analysis showed that first or second intron in type II MADS-box genes is considerably longer than the longest intron of MIKC* or M-type genes reaching to about 22 kb in *TdFLC-3A-1* (Fig 3A). The mean number of exons in *T. dicoccoides* MADS-box genes was 1.29 (in M-type genes), 6.47 for MIKC-type genes, and 9.67 in MIKC*-type genes (Fig 3B). However, MIKC$^C$-type genes were significantly longer than the M- and MIKC* type genes: the mean gene length was 1.00 kb in M-type genes, 10.53 kb in MIKC$^C$-type genes, and 3.07 kb in MIKC* type genes (Fig 3B). MIKC*-type genes in *T. dicoccoides* have an average number of exons (9.67) almost equal to that of Arabidopsis (10).

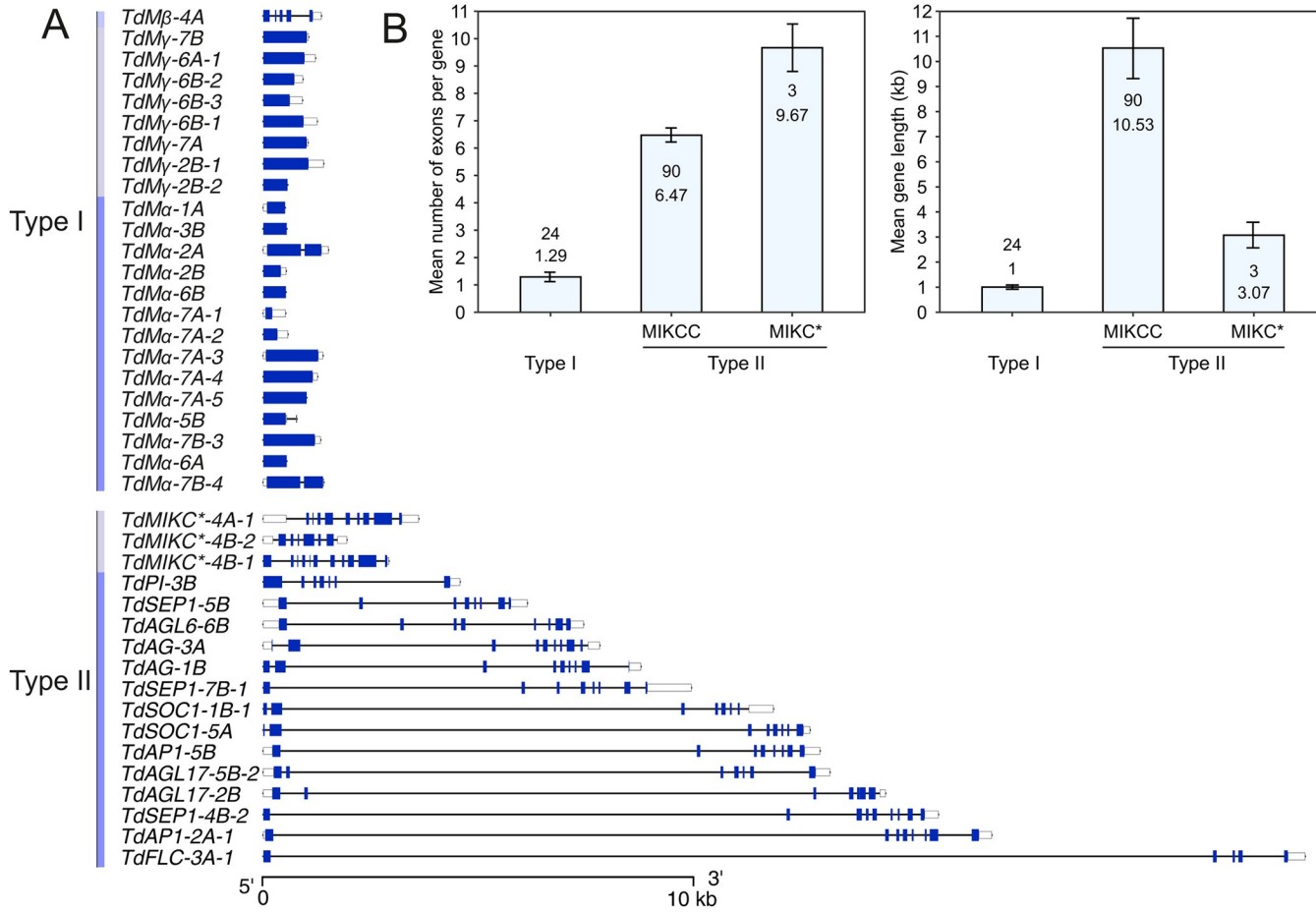

**Fig 3. The structure of MADS-box genes in *T. dicoccoides*.** A) Comparison of exon-intron structures between type I and a representative sample of type II genes. Exons are in blue; 3' and 5' untranslated regions (UTRs) are shown in white and introns are represented by black lines. B) The mean number of exons and the mean length of MADS-box genes (± standard errors). The number of genes in each group is also indicated. Almost all type I MADS-box genes were single exon genes.

## Phylogenetic analysis and distribution of MADS-box genes

Based on a maximum likelihood phylogenetic analysis of MADS-box genes from wild emmer wheat, rice, and Arabidopsis, 17 main grass subfamilies of MADS-box gene including *SOC1* (*SUPPRESSOR OF OVEREXPRESSION OF CONSTANS1*), *AG/STK* (*AGAMOUS/SEED-STICK*), *SEP1* and SEP3 (*SEPALLATA*), *AGL6* (*AGAMOUS-LIKE6*), *AGL12*, *AP1*, *AGL17*, *Bsister*, *PI* (*PISTILLATA*), *SVP* (*SHORT VEGETATIVE PHASE*), *AP3*, *OsMADS32*, monocot and Arabidopsis *FLC* (*FLOWERING LOCUS C*) groups, *Mγ*, *Mβ*, and *Mα* [69, 70] were identified in wild emmer wheat. The rice and wild emmer wheat *FLC* clade composed a district clade different from the Arabidopsis *FLC* clade [6, 71] and hence was called monocot *FLC* (Fig 1). The phylogenetic tree shows that *AGAMOUS*, *AGL12*, *AP1*, *SVP*, *OsMADS32*, and *MIKC** genes of Arabidopsis have conserved sister groups in wild emmer wheat, however, some *SOC1*, *SEP1*, *AGL17*, *FLC* individuals in wild emmer wheat have gained additional copies (7:2 wild emmer wheat to rice copies for *SOC1*, 10:3 for *SEP1*, 14:5 for *AGL17* and 14:2 for monocot *FLC*) probably due to duplication events during evolution. On the other hand, the number of M-type MADS-box genes in wild emmer wheat was considerably lower than that of Arabidopsis (24:56), especially only one distantly related Mβ was found in wild emmer wheat compared to 19 orthologous copies of Arabidopsis (Fig 1).

*T. dicoccoides* contain almost two-fold MIKC type MADS-box genes (93) than Arabidopsis with 45 MIKC-type genes [24]. When considering the number of MIKC-type genes per subgenome, it seems that this significantly higher number is mainly the result of polyploidy because the number of MIKC-type MADS-box genes per subgenome in wild emmer (with the two A and B subgenomes) is 93/2 = 46.5 which is similar to that of rice with 43 and Arabidopsis with 45 Type II MADS-box genes. On the other hand, the number of M-type genes in wild emmer (27) is lower than those of Arabidopsis (62) [24] and rice (32) [49]. None of the M-type MADS-box genes of wild emmer wheat contain K domain and most of the type-I MADS-box genes in wild emmer wheat show zero or very low expression compared to their type II homologs (S2 Fig).

Wild emmer wheat contains 14 *AGL17*-like genes, which is more than two-fold of the six *AGL17*-like genes in rice genome (Fig 1). On the other hand, this number is reasonably lower than two-thirds of the number of common wheat where 47 *AGL17* members have been identified [6]. A two-third ratio is expected in gene number of wild emmer wheat containing A and B subgenomes compared to common wheat containing A, B and D subgenomes. It seems that the higher number of *AGL17* genes in common wheat is the result of their tandem duplications mainly on chromosome 7, resulting the skewed common wheat-to-emmer wheat gene ratio of the *AGL17* genes. Five of the *AGL17* members in *T. dicoccoides* encode both MADS- and K-domain (Fig 1, green dots in *AGL17* clade), and the other nine genes only encode a MADS domain (Fig 1, red dots in *AGK17* clade). *T. dicoccoides* has 16 *FLC* members (Fig 1, monocot *FLC* clade; Fig 2), which is noticeably higher than the two *FLC* genes from rice. Most of wheat *FLC*-like genes (8 out of 14) were located on the long arm of homoeologous group 3 in close vicinity to each other, suggesting the involvement of tandem duplication.

The rice genome contains three *Bsister* paralogs including *OsMADS29*, *OsMADS30*, and *OsMADS31*. In wild emmer wheat, *OsMADS29*-like and *OsMADS31*-like genes were present in syntenic locations in both A and B subgenomes but *OsMADS30* only had one ortholog in wild emmer wheat which was located on B subgenome (Fig 1). All these five *Bsister* members in wild emmer, contained both MADS and K coding domains suggesting retention of a conserved structure and function. A wider spread for *Bsister* members has already been found in common wheat where 27 conserved or truncated *OsMADS30* homologs were dispersedly located in different chromosomes [6]. Such a dispersed distribution homologous genes might

be the result of transposon activity by capturing full or partial gene sequences and transpose them to another location.

In *SEP* clade, two *SEP3* members out of four rice orthologs were assigned to a pair of *T. dicoccoides* homoeologs, resulting in the expected 1:2 ratio. However, the *SEP1* rice genes (i.e. *OsMADS1* and *OsMADS5*) were grouped with 5 (2 + 3) and 3 (2 + 1) wild emmer wheat genes on chromosomes 4 and 7, respectively (Fig 2), suggesting occurrence of gene duplications in wild emmer wheat *SEP1* subclades.

## Cis-acting elements in TdMADS-box promoters

To better understand how *T. dicoccoides* MADS-box genes regulate external stimuli, the promoter regions of the 117 *T. dicoccoides* MADS-box genes were analyzed using the PlantCARE database. The analysis detected 3135 cis-acting elements possibly responding to light, hormones, stress, endosperm meristem, etc. (Table 1 and Fig 4). Each of the TdMADS-box genes contained 6 to 48 responsive elements, mainly related to light, however hormone, drought, and low-temperature related cis-acting elements were also present. Promoter analysis further showed that TdMADS-box genes might also be involved in responses to methyl jasmonate (MeJA), ABA, auxin, gibberellin, and salicylic acid. Overall, the results suggest that the TdMADS-box family members generally respond to light and could play a role in hormone responses and abiotic stresses.

## miRNAs target analysis

With stringent cut-off expectation threshold of $\leq 0.5$, psRNATarget [68] detected nine MADS-box cDNA target candidates in wild emmer wheat genome. All these cDNA sequences are predicted to be the target for miR444. Two target sites were predicted for the cDNA of each of the *TdAGL17-6B* and *TdAGL17-6A* genes while each of the remaining cDNAs contained only one target site. Furthermore, 41 different miRNA-target sequences were identified on intron sequences of 32 TdMADS-box genes at the same expectation value of 0.5 (S3 Table). At the intron level, some genes for example *TdSOC1-1A-1*, *TdSOC1-1A*, *TdAP3-7A*, *TdAP3-*

**Table 1. Cis-acting elements on the promoter region of MADS-box genes in *T. dicoccoides*.** Overall cis-acting elements on the 2kb upstream of MADS-box genes related to different stimuli are presented.

| Stimulus | Number of elements |
|---|---|
| Light | 1290 |
| Methyl Jasmonate (MeJA) | 585 |
| Abscisic acid | 419 |
| Auxin | 119 |
| Gibberellin | 84 |
| Salicylic acid | 54 |
| Drought | 88 |
| Low temperature | 92 |
| Defense stress | 33 |
| Anaerobic induction | 187 |
| Cell cycle | 11 |
| Circadian control | 25 |
| Endosperm expression | 22 |
| Meristem expression | 90 |
| Seed | 36 |
| Total | 3135 |

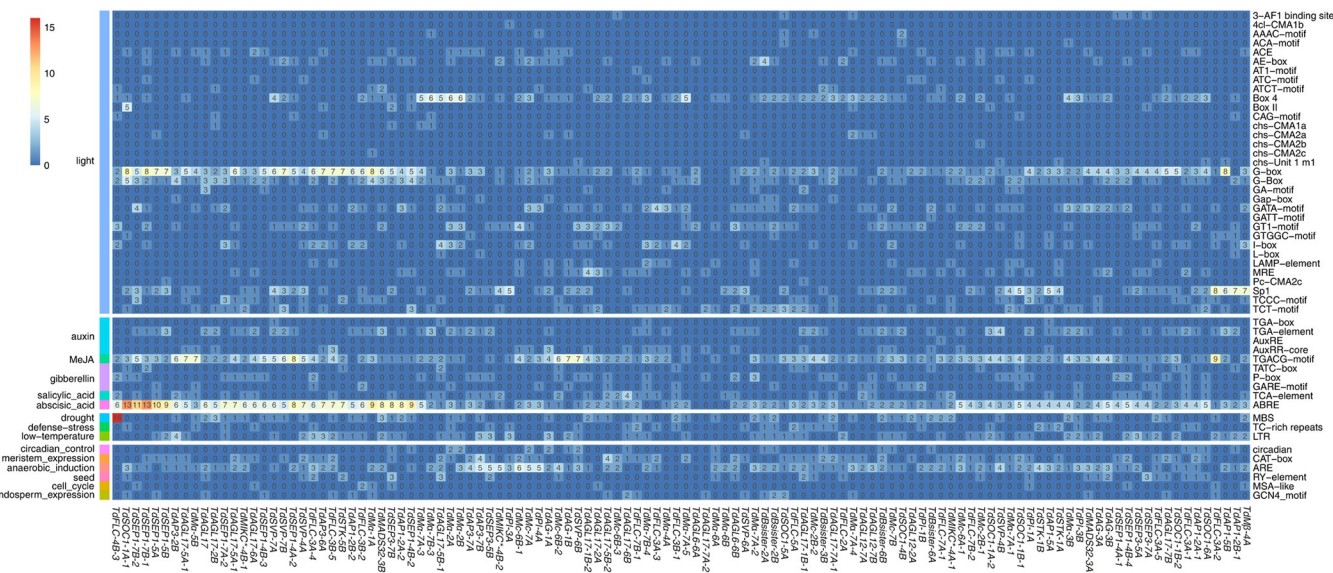

**Fig 4. Potential cis-acting elements in promoter region of TdMADS-box genes of *T. dicoccoides*.** The number of potential cis-acting elements in 2-kb upstream promoter region of TdMADS-box genes were predicted using the PlantCARE database [67]. The number of each cis-acting element (shown on the right side) identified for each gene is presented inside the cells.

## MADS-box gene expression during developmental stages

153 samples of RNA-seq data from 20 different combinations of wild emmer wheat were analyzed. The samples were from tissues and developmental stages belonging to root, leaf, flag leaf, flower (anthers and carpels), glume, lemma and palea, grain, and different stages of developing spike [60]. The resulting MADS-box gene expression values and modules are presented in S4 and S5 Tables. Out of 117 emmer wheat MADS-box genes, 85 were expressed in at least one developmental stage, with a maximum expression ranging from 1.12 to 7.97 log2 (FPKM + 1). The maximum expression rates of the remaining 32 genes varied from 0 to 1 log2 (FPKM + 1) (Fig 5 and S4 Table and S2 Fig). Most of the *AGL17* genes are expressed at zero to low rates except for the *TdAGL17-6A* and *TdAGL17-6B* which are expressed in root, vegetative and reproductive organs. AG/STK genes are mainly expressed in flower and grain, SEP3, AGL6, PI and AP3 genes are expressed in flower and/or grain and to lower extents in developing spike (Fig 5A and 5B). *T. dicoccoides* contains 5 *Bsister* copies that are mainly expressed in flower and grain. It is well known that *Bsister* genes are expressed in ovule and grain with involvement in seed development [41, 72, 73]. In total, the type II MADS-box expression patterns of wild emmer wheat are similar to those of common wheat [6] and rice [49]. M-type MADS-box genes showed zero or week expression levels in wild emmer. Out of 16 M-type TdMADS-box genes, 14 expressed only in grain (123 days from sowing) at a maximum expression rate of 1.25 log2 (FPKM + 1). *TdMβ-4A* and *TdMγ-6B-3* were also expressed in flowers (S4 Table and S2 Fig).

Nine different expression modules were detected following co-expression analysis of the MADS-box gene. The expression patterns in the resulting modules (Fig 5B) generally showed similarity to the expression of MADS-box genes subfamilies (Fig 5A). For example, AG/STK members were grouped into two adjacent modules. Similarly, most M-type genes were

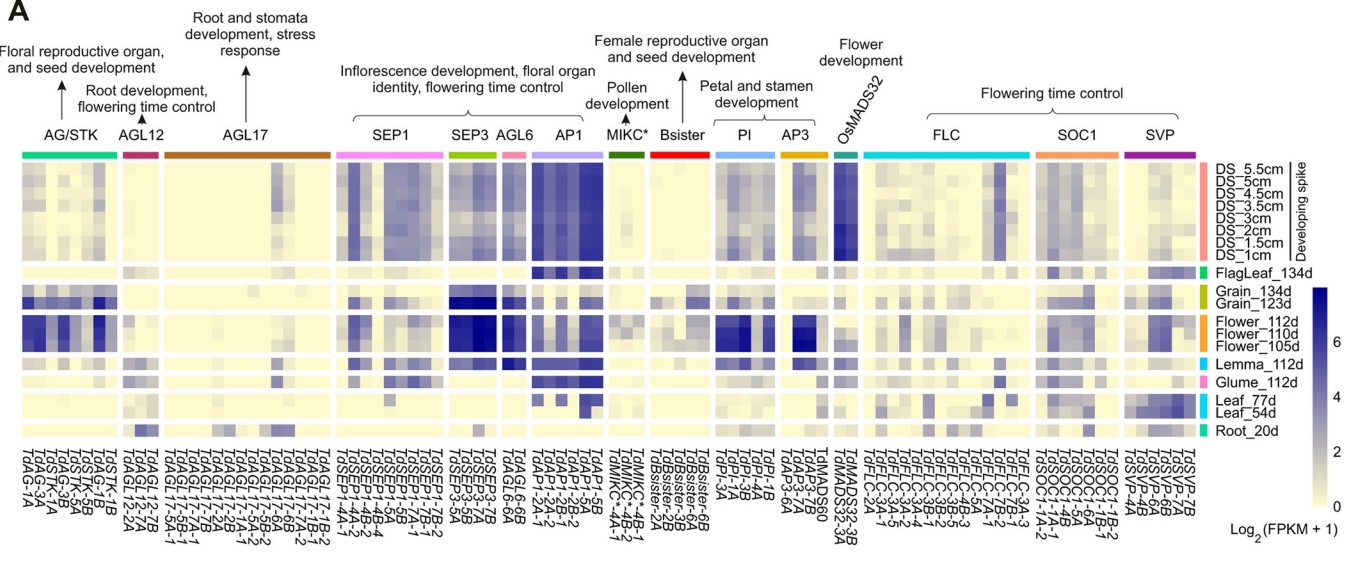

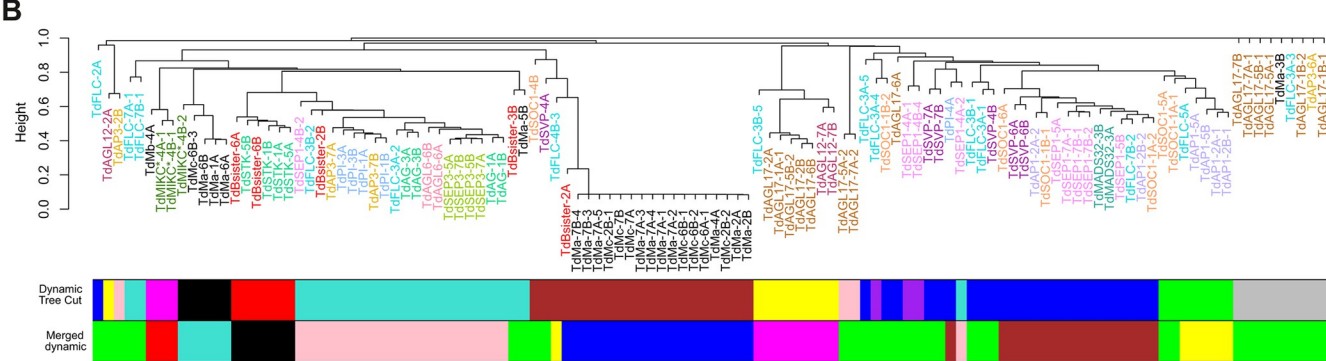

**Fig 5. Expression analysis of the *T. dicoccoides* MADS-box genes as reveled by RNA-seq data.** A) A heat map of mean expression of type-II MADS-box genes in different tissues and developmental stages of *T. dicoccoides*. Numbers followed by the developing stages are days from sowing (d) or spike length (cm). B) Co-expression clustering of the *T. dicoccoides* MADS-box genes based on their expression values from different tissues and developmental stages. Colors indicate the different modules. Note that M-type MADS-box genes were not presented in 'A' but they were included in co-expression pattern analysis in 'B'.

grouped into a single module (Fig 5B) which indicated no or very low expression pattern (S4 Table). Genes from some subfamilies showed considerable differences in their expression patterns. For example, members of FLC and SEP subfamilies have been located in different modules.

From the microarray data, 7 differentially expressed MADS-box genes were found under drought stress in at least one of the two evaluated genotypes, among which, only *TdSOC1-6A* upregulated in both drought susceptible and drought tolerant genotypes under drought conditions while *TdPI-1A*, *TdSOC1-1A-1*, *TdSOC1-6A* and *TdAGL12-7A* differentially expressed only in the tolerant genotype (Fig 6 and S6 Table).

## Discussion

### The conserved function of TdMADS-box genes

MADS-box transcription factors play important roles in various processes of plant development, such as floral organ identity determination, flower development, and seed formation.

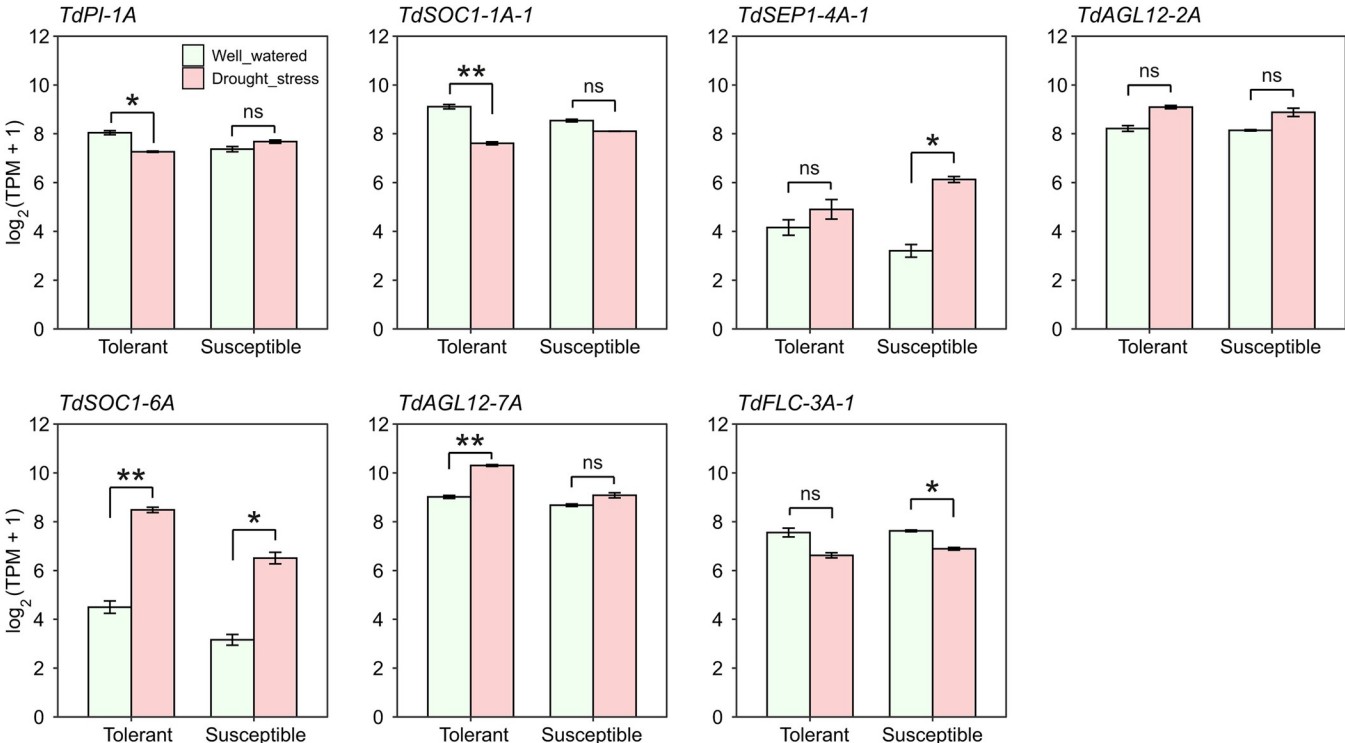

**Fig 6. Expression of MADS-box genes of wild emmer wheat under drought stress.** Mean expression (± standard error) of MADS-box genes of wild emmer wheat which differentially responded to drought stress conditions as revealed by microarray data (GEO accession: GSE31762). The microarray data belongs to the flag leaf of two wild emmer wheat genotypes contrasting in their productivity and yield stability under terminal drought stress.

They are also involved in responding to environmental stresses. Here, I identified 117 MADS-box genes in the wild emmer wheat genome which is an important source for wheat improvement. Phylogenetic analysis along with the MADS-box genes of rice and Arabidopsis assigned emmer wheat MADS-box genes to 17 (14 MIKC-type and 3 M-type) subfamilies (Fig 1 and S1 Table).

In general, a high similarity in the expression pattern between wild emmer wheat MADS-box genes, common wheat [6, 74] and rice [49] orthologs was found indicating a conserved functionality of MADS-box genes between these species. In wild emmer wheat, *TRID-C5AG057030* (named *TdAP1-5A*) and *TRIDC5BG061170* (named *TdAP1-5B*) are vernalization *VRN-A1* and *VRN-B1* genes respectively. *TRIDC2AG022240* (*TdAP1-2A-1*) and *TRIDC2BG025920* (*TdAP1-2B-1*) of wild emmer wheat is co-expressed with *Vrn1-5A* (Fig 5A and S4 Table), which indicates that this gene may also be related to flowering time. Similarly, in common wheat, *TRAESCS2D02G181400* which is orthologous to *TRIDC2AG022240* (*TdAP1-2A-1*) and *TRIDC2BG025920* (*TdAP1-2B-1*) of wild emmer wheat encodes a MIKC-type MADS-box transcription factor and is co-expressed with *Vrn1-5A* [75]. Different alleles and copy number variation of *VRN* genes involve in the transition of the shoot apical meristem to the reproductive phase [76–78]. The spring forms of emmer wheat are associated with the independent emergence of a new dominant *VRN-A1* allele which resulted from changes in the promoter region and a large deletion in the first intron [77, 79]. The wild-type *VRN1* allele for winter growth habit requires long exposures to low temperatures (vernalization) to be expressed, so *VRN1* has a pivotal role in the determination of flowering time.

Interestingly, the number of M-type genes in wild emmer wheat (27) is significantly lower than that of Arabidopsis (62) [24]. None of the M-type genes contained K domain. Truncated

genes are common among M-type genes also they may be functional. In *A. thaliana* some M-type MADS-box genes are important for normal development of the female gametophyte or endosperm and may be responsible for post-zygotic lethality in interspecific hybrids [25–31]. Out of 16 M-type TdMADS-box genes of wild emmer wheat, 14 were expressed, albeit at very low rates and mostly only in grain (123 days from sowing) with a maximum expression rate of 1.25 log2 (FPKM + 1). *TdMβ-4A* and *TdMγ-6B-3* were also expressed in flower (S4 Table and S2 Fig). Similarly in common wheat, almost 75% of non-expressed MADS-box genes were members of the type I clade [4]. In agreement with the results obtained here, Nam, Kim [33] found a higher proportion of nonfunctional genes among the type I MADS-box group and suggested that type I genes have undergone a higher rate of birth-and-death evolution than type II genes in angiosperms which might be the result of more frequent segmental duplications and less purifying selection of type I than in type II genes [33].

## Gene duplication versus environmental stresses

Tandem duplicates may be correlated to the adaptation to different environments [80]. Duplications of large chromosomal segments i.e. segmental duplications in most cases appear to have come from one round of polyploidy [81]. *T. dicoccoides* is the oldest polyploid wheat and the ancestral species of common wheat. By comparing to diploid and hexaploid *Triticum* species, it provides an opportunity to study the MADS-box gene family members during polyploidization. Contrary to common wheat which has undergone extensive expansion of some MIKC-type subfamilies [4, 6], the number of MIKC-type MADS-box genes per subgenome in wild emmer wheat is generally comparable to that of rice and Arabidopsis. The number of MIKC-type MADS-box genes per subgenome in wild emmer is 93/2 = 46.5 compared to that of rice (43) and Arabidopsis (45) Type II MADS-box genes. However, some subfamilies including *SOC1*, *SEP1*, *AGL17*, and *FLC* showed moderate to high rates of duplication per subgenome compared to the rice genome (7:2 copy ratio for SOC1, 10:3 for *SEP1*, 14:5 for *AGL17* and 14:2 for *FLC*). It has been suggested that the expansion of eudicot *FLC* genes potentially enables the ability to adapt to various environmental conditions including ambient temperatures [82]. The high level of duplication of *FLC* genes in *T. dicoccoides* may similarly contribute to its adaptation to different environments by altering its flowering time [12]. *FLC* plays a crucial role in regulating the flowering time in plants. *FLC* represses flowering transition by repressing promoters of flowering genes, such as *FT* and *SOC1*. During vernalization, *FLC* protein levels decrease and therefore flowering is induced [83–85]. The analysis showed that the Arabidopsis *FLC* clade composed a district clade different from the rice and emmer wheat (monocot) FLC clade (Fig 1) [6, 71]. The presence of *FLC-like* genes in cereals was unknown for a long time, even though there was a lot of information about Arabidopsis *FLC*. It has been suggested that mechanisms developmental and flowering time regulation in monocots compared to eudicots and thought that that *FLC* only existed in eudicot plants [Reviewed in 86]. But the synteny analysis and phylogeny has been proven that *FLC* relatives are presence in cereals which are related to the *FLC* genes of Arabidopsis [71]. There are two subclades within this monocot *FLC* group, called the *OsMADS51* and *OsMADS37* subclades as these rice genes were located within each group (Fig 1).

## Some MADS-box genes are expressed in response to drought stress

I observed an upregulation in response to stresses for some MIKC-type MADS-box genes. Specifically, *TdPI-1A*, *TdSOC1-1A-1*, *TdSEP1-4A-1*, *TdSOC1-6A*, *TdAGL-12-7A*, and *TdFLC-3A-1* differentially responded to drought stress condition as revealed by microarray data (Fig 6). Studies have shown that some MADS-box genes such as *AGL12* and *MBP8* have a negative

role in drought [11–13] and cold [14] tolerance by regulating the expression of genes involved in stress response pathways. In rice, overexpression of the *TdAGL12-7A* ortholog (i.e. *MADS26*) is possibly connected to response to stresses.

OsMADS26-down-regulated plants also have shown enhanced resistance against two rice pathogens. In spite of this improved resistance under biotic stresses, *OsMADS26*-down-regulated plants also showed more tolerance to drought stress in both controlled and field conditions without a strong impact on plant development [15]. Other MADS-box genes might also be involved in abiotic stress in plants. For example, it has been shown that nitrate-dependent salt tolerance is mediated by *OsMADS27* in rice (orthologous to *TdAGL17-2A* and *TdAGL17-2B* of wild emmer wheat) where the expression of *OsMADS27* was specifically induced by nitrate [16]. In Arabidopsis, *SVP* is also a major regulator of ABA catabolism and *SVP*, *CYP707A1/3*, and *AtBG1* together are involved in plant response to water stress and plant drought resistance [17].

## Cis-regulatory elements and introns of MADS-box genes may contribute to environmental adaptation

The analysis of cis-acting elements in TdMADS-box promoters suggests that the TdMADS-box family generally responds to light and could play a role in hormone responses and abiotic stresses. Mutations in the *VRN1* (*AP1*) promoter region or deletions in its first intron results in a spring growth habit as the vernalization in not required for flowering [87]. Insertion of a GATA box like sequence at the promoter region of the *VRN-A3* locus in a cultivated emmer wheat genotype (*Triticum turgidum* L. ssp. *dicoccum*) confers early flowering trait [88].

miRNAs are 20–24 nucleotides in size and promote degradation or repression of translation of target mRNAs, herby negatively regulate gene expression at post-transcriptional level. miR444 is a monocot-specific microRNA. It has been shown that miR444 is a key factor for virus resistance via RNA-silencing in rice. miR444 reduces the repressive roles of *OsMADS23*, *OsMADS27a*, and *OsMADS57* on *OsRDR1* transcription, thus the OsRDR1-dependent antiviral RNA-silencing pathway is activated [89]. Similarly, miR444 also plays a role in rice tillering [90]. miR444 and its target *OsMADS27* TF are also involved in NO3-dependent root development [91]. NO3$^{-}$ depression induces miR444 expression, and the expression of a miR444 target can quench the miRNA and act as a sponge in transgenic rice lines resulting in increased total root growth [92]. At the intron level, some genes for example *TdSOC1-1A-1*, *TdSOC1-1A*, *TdAP3-7A*, *TdAP3-2B*, *TdAGL17-2A*, *TdSOC1-1A-2*, *TdSTK-5B*, *TdAGL17-6B*, *TdSVP-4B* and *TdSEP1-4A* contained different miRNA-target sequences. Intronic miRNAs are transcribed from introns of protein-coding genes. They have been shown to be involved in post-transcriptional regulation of gene expression [93]. Furthermore, intron sequences can form circular RNA. circular RNA containing miRNA sequences can regulate the expression of mRNAs by acting as miRNA sponges as well [94]. It has been shown that the miRNA444 is upregulated in *T. aestivum* under salt stress. Similarly, the miR1120 which was identified on the intron of *TdAGL17-2A* and *TdSOC1-1A-1* is upregulated under salt stress in *T. dicoccoides* [95].

## Conclusion

There are evidences about the involvement of the MADS-box genes in plant growth, development and stress tolerance, however, the detailed information on MADS-box gene family was not available in wild emmer wheat. Here, a genome-wide analysis showed that the MADS-box genes in wild emmer wheat especially MIKC-type clades have retained conserved functionality. MADS-box genes in wild emmer wheat have promoters responsive to various stimuli and play important roles in growth and development and response to stresses. The specific

adjustment via gene duplication, the alterations in expression patterns under various conditions such as photoperiod, temperature, and stresses, and promoter and intronic sequence evolution have all contributed to fine-tuning the MADS-box gene functionality. The results provide comprehensive information about the MADS-box genes of wild emmer wheat that could accelerate functional genomics efforts and potentially facilitate bridging gaps toward breeding for agronomically important traits in wheat.

## Supporting information

**S1 Fig. Domain structure of *T. dicoccoides* MADS-box proteins.** Conserved domain predictions for the 117 *T. dicoccoides* MADS-box proteins are presented.
(PDF)

**S2 Fig. Expression heatmap of *T. dicoccoides* MADS-box genes.** The heat map represents mean expression of 117 *T. dicoccoides* MADS-box genes in different tissues and developmental stages.
(PDF)

**S1 Table. The MADS-box gene description and protein sequences.** The table contains MADS-box gene description and protein sequences of *T. dicoccoides*, *Oryza sativa*, and *Arabidopsis thaliana*.
(XLSX)

**S2 Table. Exon, intron, and CDS boundaries of the *T. dicoccoides* MADS-box genes.**
(XLSX)

**S3 Table. Predicted miRNAs targeting MADS-box gene family in *T. dicoccoides*.**
(XLSX)

**S4 Table. Expression data of *T. dicoccoides* MADS-box genes.** *T. dicoccoides* MADS-box gene expression data based on log2(FPKM + 1) in different tissues and developmental stages.
(XLSX)

**S5 Table. Expression module data for *T. dicoccoides* MADS-box genes related to Fig 5.**
(XLSX)

**S6 Table. Expression data of *T. dicoccoides* MADS-box genes.** The expression data of *T. dicoccoides* MADS-box gene which differentially responded to drought stress conditions as reveled by microarray data.
(XLSX)

## Author Contributions

**Conceptualization:** Ghader Mirzaghaderi.

**Formal analysis:** Ghader Mirzaghaderi.

**Funding acquisition:** Ghader Mirzaghaderi.

**Investigation:** Ghader Mirzaghaderi.

**Methodology:** Ghader Mirzaghaderi.

**Writing – original draft:** Ghader Mirzaghaderi.

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
