## [Decision Letter · Decision Letter 0]

13 Feb 2024

PONE-D-23-35009Genome-wide analysis of MADS-box transcription factor gene family in wild emmer wheat (Triticum turgidum subsp. dicoccoides)PLOS ONE

Dear Dr. Mirzaghaderi,

Thank you for submitting your manuscript to PLOS ONE. After careful consideration, we feel that it has merit but does not fully meet PLOS ONE’s publication criteria as it currently stands. Therefore, we invite you to submit a revised version of the manuscript that addresses the points raised during the review process.

We look forward to receiving your revised manuscript.

Kind regards,

Muhammad Abdul Rehman Rashid, PhD

Academic Editor

PLOS ONE

 [GM was supported by Iran National Science Foundation (INSF) grant 99014038. The funding doesn't include publication fee.].  

[GM was supported by Iran National Science Foundation (INSF) grant 99014038.]

 [GM was supported by Iran National Science Foundation (INSF) grant 99014038. The funding doesn't include publication fee.]

5. Please amend your authorship list in your manuscript file to include author Ghader Mirzaghaderi.

Reviewers' comments:

Reviewer's Responses to Questions

**Comments to the Author**

1. Is the manuscript technically sound, and do the data support the conclusions?

Reviewer #1: Yes

Reviewer #2: Yes

2. Has the statistical analysis been performed appropriately and rigorously? 

Reviewer #1: Yes

Reviewer #2: Yes

3. Have the authors made all data underlying the findings in their manuscript fully available?

Reviewer #1: Yes

Reviewer #2: Yes

4. Is the manuscript presented in an intelligible fashion and written in standard English?

Reviewer #1: Yes

Reviewer #2: Yes

5. Review Comments to the Author

Reviewer #1: In the manuscript (Genome-wide analysis of MADS-box transcription factor gene family in wild emmer wheat), The authors demonstrated that members of MADS-box gene family played an important role in the regulation of plant growth and development. A total of 117 MADS-box genes were identified in the wild emmer wheat genome, which were classified into MIKCC, MIKC and 2M types. The gene expression profiles in different tissues and developmental stages conform to the MADS-box gene subfamily classification, which is similar to common wheat and rice, suggesting that they have conserved functions. Some MADS-box genes were differentially expressed under drought stress. The response elements in the promoter region of the MADS-box gene are mainly related to light sources, but are also related to hormones, drought and low temperature. The results revealed detailed information about the MADS-box gene in wild emmer wheat. However, I think the manuscript should be revised before publication.

Major:

1. Articles cited need to be time-sensitive, the years of some published references are too long, try to refer to the literature of the past ten years.

2. The introduction part need to be polished further to highlight the background and importance of the manuscript.

3. Some of the language in the manuscript needs improvement. Some sentences should be reorganized.

4. The order of the materials and methods is somewhat chaotic, and should be elaborated in sequence according to the primary and secondary order.

5. The first paragraph of the discussion section does not highlight the important themes of the article, and the discussion needs to better clarify the importance of the current findings and how they take the field forward.

6. As an important part of the paper, the conclusion is too simple and needs to be enriched.

Minor:

1. Page 5, the part "Naming of MIKC-type MADS-box genes" should be placed in the first part of "Materials and Methods".

2. Page 12, Line 17, "Treas", should be capitalized as "TRAES".

3. Page 14, Line 6, the background color of "presence" is incorrectly used.

4. Some of the formats of the references need to be adjusted.

Reviewer #2: The purpose of the study seems somewhat sound but some major revisions need to be made before it can proceed further for proper consideration in PLoS One.

1. The ‘Introduction’ section should begin with a background to wheat and the problems its facing and then the significance of the gene family in other wheat species (in any available) or closely related species.

2. Give the ‘e-value’ for BLASTp used with HMM. Also, the reason behind using old MAFFT instead of MUSCLE as it is a medium-to-large protein multiple sequence alignment.

3. Abbreviation detail should only be given when first used. Check this throughout the manuscript.

4. Experimental verification of curated and potent MADS-box genes through qRT-PCR should be provided to enhance the quality of the results and the overall study in study.

5. The results are also not clearly written with some instances of using first person singulars (such as under the heading ‘Expression of MADS-box genes’). This should be rectified.

6. Although, the overall write-up of the manuscript is descent yet I would suggest the use of suitable punctuation (such as in abstract), break large sentences into smaller ones, italicize the scientific names and genes, and re-read the manuscript for clarity.

6. PLOS authors have the option to publish the peer review history of their article (what does this mean?). If published, this will include your full peer review and any attached files.

Reviewer #1: **Yes: **Guang-Long Wang

Reviewer #2: No

---

## [Author Response · Author response to Decision Letter 0]

15 Feb 2024

Dear Editor,

I greatly appreciate your attention and valuable suggestions to may manuscript. I applied the constructive and valuable comments of you and the reviewers as follow. The figures format was also adjusted using the https://pacev2.apexcovantage.com/ website. I hope that the applied changes and corrections will meet you and the reviewer’s approval and consent.

I updated the financial disclosure as follow:

Funding 

GM was supported by Iran National Science Foundation (INSF) grant 99014038. The funders had no role in study design, data collection and analysis, decision to publish, or preparation of the manuscript.

Response to reviewers:

Reviewer #1:

In the manuscript (Genome-wide analysis of MADS-box transcription factor gene family in wild emmer wheat), The authors demonstrated that members of MADS-box gene family played an important role in the regulation of plant growth and development. A total of 117 MADS-box genes were identified in the wild emmer wheat genome, which were classified into MIKCC, MIKC and 2M types. The gene expression profiles in different tissues and developmental stages conform to the MADS-box gene subfamily classification, which is similar to common wheat and rice, suggesting that they have conserved functions. Some MADS-box genes were differentially expressed under drought stress. The response elements in the promoter region of the MADS-box gene are mainly related to light sources, but are also related to hormones, drought and low temperature. The results revealed detailed information about the MADS-box gene in wild emmer wheat. However, I think the manuscript should be revised before publication.

Major:

1. Articles cited need to be time-sensitive, the years of some published references are too long, try to refer to the literature of the past ten years.

The text, especially the Introduction was updated with more recent publications. Although some old papers are basis in MADS-box studies.

2. The introduction part needs to be polished further to highlight the background and importance of the manuscript.

Thanks. The introduction was undated as mentioned.

3. Some of the language in the manuscript needs improvement. Some sentences should be reorganized.

I went through the manuscript and tried to improve the sentences.

4. The order of the materials and methods is somewhat chaotic, and should be elaborated in sequence according to the primary and secondary order.

Thanks. Applied. As you suggested, by moving “Naming of MIKC-type MADS-box genes” to after “Identification of MADS-box genes in T. dicoccoides”, I think it is now more appropriate.

5. The first paragraph of the discussion section does not highlight the important themes of the article, and the discussion needs to better clarify the importance of the current findings and how they take the field forward.

Yes, changed to highlight the importance of MADS-box genes in wheat.

6. As an important part of the paper, the conclusion is too simple and needs to be enriched.

The Conclusion was updated.

Minor:

1. Page 5, the part "Naming of MIKC-type MADS-box genes" should be placed in the first part of "Materials and Methods".

Thanks. Applied. It was moved to after Identification of MADS-box genes in T. dicoccoides.

2. Page 12, Line 17, "Treas", should be capitalized as "TRAES".

Thanks. Corrected.

3. Page 14, Line 6, the background color of "presence" is incorrectly used.

Corrected.

4. Some of the formats of the references need to be adjusted.

I went trough references and made corrections where required.

Reviewer #2: 

The purpose of the study seems somewhat sound but some major revisions need to be made before it can proceed further for proper consideration in PLoS One.

1. The ‘Introduction’ section should begin with a background to wheat and the problems its facing and then the significance of the gene family in other wheat species (in any available) or closely related species.

Thanks. The introduction was undated as mentioned and wheat and the facing problems as well as gene family significance were provided first.

2. Give the ‘e-value’ for BLASTp used with HMM. Also, the reason behind using old MAFFT instead of MUSCLE as it is a medium-to-large protein multiple sequence alignment.

Added: …. at the 0.001 p-value cut-off…

I agree that MUSCLE is powerful protein aligner. I also tried the MUSCLE but the resulting tree was more reasonable when made using MAFFT. MAFFT is a motif-based sequence aligner and matches the conserved domains better when the intervening segments are variable between different sequences. I think it is because of the introduced new feature for suppressing over-alignment (aligning unrelated segments). Ref: https://academic.oup.com/bioinformatics/article/32/13/1933/1743504

3. Abbreviation detail should only be given when first used. Check this throughout the manuscript.

Checked and corrected. Many thanks.

4. Experimental verification of curated and potent MADS-box genes through qRT-PCR should be provided to enhance the quality of the results and the overall study in study.

Unfortunately, the qRT-PCR currently is not possible for me. I hope that the current data of the manuscript will meet your consent. 

5. The results are also not clearly written with some instances of using first person singulars (such as under the heading ‘Expression of MADS-box genes’). This should be rectified.

I went through the manuscript and tried to improve the writing.

6. Although, the overall write-up of the manuscript is descent yet I would suggest the use of suitable punctuation (such as in abstract), break large sentences into smaller ones, italicize the scientific names and genes, and re-read the manuscript for clarity.

I went through the manuscript and tried to improve the sentences.

Many thanks,

Best regards,

Ghader Mirzaghaderi

---

## [Editor Report · Decision Letter 1]

20 Feb 2024

Genome-wide analysis of MADS-box transcription factor gene family in wild emmer wheat (Triticum turgidum subsp. dicoccoides)

PONE-D-23-35009R1

Dear Dr. Mirzaghaderi,

We’re pleased to inform you that your manuscript has been judged scientifically suitable for publication and will be formally accepted for publication once it meets all outstanding technical requirements.

Kind regards,

Muhammad Abdul Rehman Rashid, PhD

Academic Editor

PLOS ONE
---

## [Editor Report · Acceptance letter]

26 Feb 2024

PONE-D-23-35009R1 

PLOS ONE

Dear Dr. Mirzaghaderi, 

I'm pleased to inform you that your manuscript has been deemed suitable for publication in PLOS ONE. Congratulations! Your manuscript is now being handed over to our production team.

Kind regards, 

on behalf of

Dr. Muhammad Abdul Rehman Rashid 

Academic Editor

PLOS ONE